# Search for ambient superconductivity in the Lu-N-H system

Pedro P. Ferreira [1,2], Lewis J. Conway [3,4], Alessio Cucciari [5,6], Simone Di Cataldo [5,7], Federico Giannessi [5,6], Eva Kogler [2], Luiz T. F. Eleno [1], Chris J. Pickard [3,4] ✉, Christoph Heil [2] ✉ & Lilia Boeri [5,6] ✉

Motivated by the recent report of room-temperature superconductivity at near-ambient pressure in N-doped lutetium hydride, we performed a comprehensive, detailed study of the phase diagram of the Lu−N−H system, looking for superconducting phases. We combined ab initio crystal structure prediction with ephemeral data-derived interatomic potentials to sample over 200,000 different structures. Out of the more than 150 structures predicted to be metastable within ~50 meV from the convex hull we identify 52 viable candidates for conventional superconductivity, for which we computed their superconducting properties from Density Functional Perturbation Theory. Although for some of these structures we do predict a finite superconducting $T_c$, none is even remotely compatible with room-temperature superconductivity as reported by Dasenbrock et al. Our work joins the broader community effort that has followed the report of near-ambient superconductivity, confirming beyond reasonable doubt that no conventional mechanism can explain the reported $T_c$ in Lu−N−H.

The report of superconductivity (SC) at near-ambient conditions in N-doped lutetium hydrides[1] raised the hope that the century-old quest for room-temperature, ambient pressure SC was finally successful[2]. A significant leap forward occurred when Mikhail Eremets' group reported conventional SC with a $T_c$ of 203 K in compressed sulfur hydride eight years ago[3]. The ensuing *hydride rush* led to the discovery of dozens of new superconductors with $T_c$'s exceeding 100 K in less than five years[4–12], thanks to an unprecedented synergy between theoretical ab initio methods and experimental investigations[13,14].

Unfortunately, the extreme pressures required to synthesize superhydrides (~Megabar) ultimately undermine the advantages of high-$T_c$ for real-world applications. In the last two years, different routes have been proposed to reduce stabilization pressures, such as optimized chemical precompression and impurity doping in ternary hydrides[15–19].

The highest $T_c$ predicted is comparable to the best predictions for non-hydride conventional superconductors ($T_c \lesssim 120$ K)[20–24]. These values are sufficient for many applications that would benefit from cost-effective liquid nitrogen cooling, which requires temperature well below the *holy-grail* limit of room-temperature SC.

Theoretically, no fundamental argument prevents room-temperature SC at ambient pressure, even within the conventional electron–phonon (*el-ph*) pairing scenario. Superhydrides have, in fact, disproven the long-held Cohen-Anderson limit for SC, showing that there are materials whose *el-ph* coupling and phonon energies allow for SC at room temperature or even higher. Unfortunately, all examples known so far require Megabar synthesis pressure[3,4,25–27].

If the report of near-room temperature SC in Lu−N−H[1] is confirmed, this compound may be the first example of a still unknown class of conventional superconductors where exceptional *el-ph*

[1]Universidade de São Paulo, Escola de Engenharia de Lorena, DEMAR, 12612-550 Lorena, Brazil. [2]Institute of Theoretical and Computational Physics, Graz University of Technology, NAWI Graz, 8010 Graz, Austria. [3]Department of Materials Science and Metallurgy, University of Cambridge, Cambridge CB30FS, UK. [4]Advanced Institute for Materials Research, Tohoku University, Sendai 980-8577, Japan. [5]Dipartimento di Fisica, Sapienza Università di Roma, 00185 Rome, Italy. [6]Enrico Fermi Research Center, Via Panisperna 89 A, 00184 Rome, Italy. [7]Institut für Festkörperphysik, Wien University of Technology, 1040 Wien, Austria. ✉e-mail: cjp20@cam.ac.uk; christoph.heil@tugraz.at; lilia.boeri@uniroma1.it

coupling properties can be realized at near-ambient conditions, and this could initiate a *second* hydride rush.

The experimental information reported by ref.[1] is, however, insufficient to identify the exact chemical composition of the new "red matter" superconductor (as coined by the authors). The first attempts to reproduce the experimental results yielded compounds with similar X-ray and absorption spectra under pressure but no trace of SC[28–35]. In this context, first-principles calculations represent an invaluable tool for investigating the possibility that a new phase with exceptional superconducting properties may exist in the Lu−N−H system.

This work aims to provide a thorough and accurate description of the near-ambient pressure superconducting phase diagram of the ternary Lu−N−H system, combining state-of-the-art, unconstrained structural searches and linear response calculations of the *el-ph* properties. Ultimately, our results will demonstrate that the Lu−N−H system lacks the conditions to harbor ambient SC within the conventional *el-ph* scenario.

## Results and discussion
### Lu−N−H phase diagram

The near-ambient superconducting state in Lu−N−H at 294 K and 1 GPa was characterized by ref.[1] through a variety of experimental techniques, including X-ray diffraction (XRD), energy-dispersive X-ray, Raman spectroscopy, as well as magnetic susceptibility, electrical resistance, and heat-capacity measurements. The onset of SC under compression is driven by a structural phase transition, associated with a change in the color of the sample from blue to pink. $T_c$ is about 100 K at 1 GPa, where SC sets in, reaches a maximum of 300 K at 2 GPa, and drops to 200 K at 3 GPa, where another structural phase transition is observed, turning the color of the sample from pink to bright red. The XRD analysis revealed the presence of two different ternary, face-centered cubic (*fcc*) Lu-networks in nearly all samples. The main phase, compound A (as identified by the authors), was indexed to space group (SG) $Fm\bar{3}m$ with lattice constant $a = 5.033$ Å at ambient pressure and underwent a structural phase transition at 3 GPa to a lower-symmetric SG, *Immm*. The second phase (compound B) can also be indexed as $Fm\bar{3}m$, but with a substantially smaller lattice constant ($a = 4.7529$ Å).

Besides these pieces of information, the authors do not report any further details: Due to the inability of conventional spectroscopy methods to accurately measure defect densities and fractional occupancies of light elements, such as hydrogen and nitrogen, even the chemical composition of the samples is unknown. Based on a comparison with known hydrides, the superconducting phase A was tentatively assigned to a $LuH_{3-\delta}N_\epsilon$ ternary structure, while compound B to rock-salt N-doped LuH. However, later theoretical and experimental works seem to suggest that phase A should most likely be characterized as pure or N-doped $LuH_2$, which is compatible with both the XRD spectra and the reported color transition, but exhibits no SC[28,29,33,34,36,37]. Furthermore, electronic structure calculations of the colors of hydrogen-defected cubic $LuH_2$ and $LuH_3$ show a strong dependence on hydrogen content, but no evidence for SC[38].

To identify structures likely to have formed in the experiments, we computed the ternary phase diagram at ambient pressure (0 GPa) and 10 GPa. To ensure that our structural search could find any relevant (meta)stable structure, we employed two different ab initio crystal structure prediction methods independently and in parallel, *viz.*, AIRSS (ab initio random structure search[39,40]) employing ephemeral data-derived potentials (EDDPs)[41], and evolutionary algorithms as implemented in USPEX[42,43]. In total, we sampled over 200,000 structures. Afterward, the structures at each pressure were merged into a single database and relaxed with the same settings to obtain a single set of convex hulls – details in the Methods Section.

Figure 1 shows the calculated convex hull at 0 GPa. The convex hull obtained at 10 GPa is reported in Supplementary Fig. 4. Circles indicate thermodynamically stable phases which form the hull, while the squares indicate metastable phases up to 50 meV/atom above the hull, with the color of the symbols corresponding to the energy distance of each phase to the hull ($\Delta E_{hull}$).

Focusing on the thermodynamically stable structures, the ternary Lu−N−H convex hull comprises four binary and one ternary phase. The binaries – $Fm\bar{3}m$-LuN (NaCl prototype), $Fm\bar{3}m$-$LuH_2$ (CaF$_2$ prototype), $P6_3$-$LuH_3$, and $P2_13$-$NH_3$ – have been experimentally reported before[44,45] or are present in crystallographic databases[46,47]. Our searches also reveal the ground state of $LuH_3$ to be of hexagonal symmetry, similar to the rhombohedral structure presented by ref. 36. As can be appreciated in the phonon dispersion plots (see below), however, this structure is dynamically unstable at the harmonic level. The cubic phase, $Fm\bar{3}m$-$LuH_3$ (AlFe$_3$ prototype), is also dynamically unstable at ambient pressures within the harmonic approximation but was predicted to be stabilized at high pressures and to become superconducting with a $T_c$ of 12.4 K (at 122 GPa)[48]. Furthermore, some of the present authors have demonstrated recently that by including temperature and quantum anharmonic lattice effects, temperatures above 200 K can, in fact, stabilize the $Fm\bar{3}m$-$LuH_3$ phase near ambient pressures[49]. We provide the crystal structure information of all thermodynamically stable phases in Supplementary Tables 1 and 2.

$Fm\bar{3}m$-LuN, $Fm\bar{3}m$-$LuH_2$, and $Fm\bar{3}m$-$LuH_3$ all contain *fcc* Lu lattices. $Fm\bar{3}m$-LuN contains nitrogen atoms on the octahedral site and has a noticeably smaller lattice constant ($a = 4.76$ Å) than the two hydrides $Fm\bar{3}m$-$LuH_2$ and $Fm\bar{3}m$-$LuH_3$, where hydrogen occupies only tetrahedral and tetrahedral + octahedral sites, respectively ($a = {\sim}5.00$ Å).

The most stable structures all fall within the $LuH_2$–$LuH_3$–LuN tie-triangle. We identify a single ternary phase, $P2/c$-$Lu_4N_2H_5$, as thermodynamically stable at 0 GPa. The crystal structure is shown in Fig. 2 and comprises layers of octahedral Lu−N bonds, as in $Fm\bar{3}m$-LuN, and tetrahedral H−Lu bonds with interstitial H atoms in the octahedral site, similar to $Fm\bar{3}m$-$LuH_3$, but with only half of the octahedral hydrogen sites occupied. This ternary phase, however, is only weakly metallic and is predicted to have a negligibly small $T_c$ according to our calculations (see Table 1). A similar structure, with $C2/m$ symmetry, is only

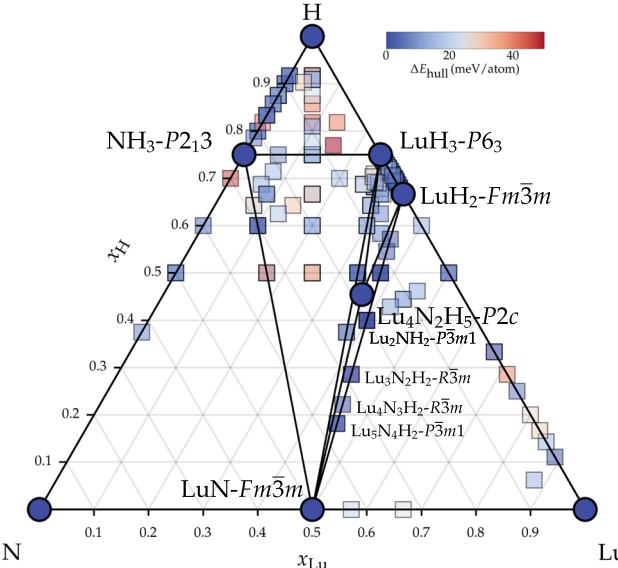

**Fig. 1 | Phase diagram of the Lu−N−H ternary system at ambient pressure.** Blue circles indicate the thermodynamically stable phases; metastable phases are shown as squares, colored according to their energy distance from the convex hull ($\Delta E_{hull}$).

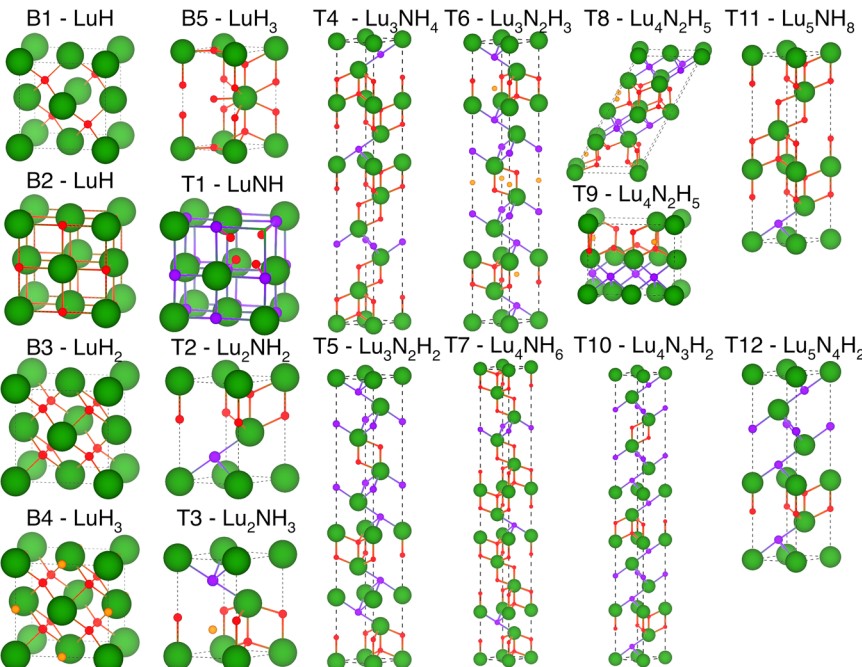

**Fig. 2 | Crystal structures of the best candidates for SC in Lu–N–H ternary system as listed in Table 1.** Lu, N, H, and H in octahedral sites are indicated as large green, medium purple, small red, and small orange spheres, respectively.

2 meV/atom higher in energy with an alternative distribution of occupied octahedral sites.

It should be noted at this point, nevertheless, that 7 ternary phases are less than 5 meV/atom distant from the hull, i.e., within the accuracy of electronic structure calculations. One also needs to consider that, depending on the system at hand and the particular synthesis route, metastable structures with enthalpies well above the convex hull can still be within reach of experimental synthesis[50]. In fact, several of these ternary structures, which, as explained above, fall within $LuH_2$–$LuH_3$–LuN tie-triangle, are thermodynamically stable at 10 GPa. (see Supplementary Fig. 4).

Considering a reasonable range of 50 meV/atom for metastability, our calculations reveal over 160 structures, with about 100 being ternary compounds, which tend to cluster along the LuN–$NH_3$, LuN–$LuH_3$, LuN–$LuH_2$, $NH_3$–$LuH_3$, and LuN–H pseudo-binary lines. More than two-thirds (108) of the metastable (binary + ternary) structures are insulating. On the H-rich side, the typical structural pattern is characterized by $H_2$, NH, $NH_2$, and $NH_3$ molecules scattered around the Lu atom or embedded in disordered motifs of Lu–N–H substructures.

Of the 52 metallic structures, i.e., the only relevant ones for SC, 32 are binaries and 20 are ternaries. In Table 1 and Fig. 2, we collected the eleven ternary metallic structures which are also dynamically stable (T2-T12). All these structures fall within the $LuH_2$–$LuH_3$–LuN tie-triangle and are structurally very similar, comprising stacking-disordered *fcc* Lu with H and N on interstitial sites. The structures on the LuN–$LuH_2$ pseudo-binary line, i.e., $P\bar{3}m1$-$Lu_2NH_2$, $R\bar{3}m$-$Lu_3NH_4$, $R\bar{3}m$-$Lu_3N_2H_2$, $R\bar{3}m$-$Lu_4N_3H_2$, $P\bar{3}m1$-$Lu_5NH_8$, and $P\bar{3}m1$-$Lu_5N_4H_2$, all comprise layers of octahedral Lu–N bonds and tetrahedral Lu–H bonds.

The structures $R\bar{3}m$-$Lu_3N_2H_3$ and $P\bar{3}m1$-$Lu_2NH_3$ correspond to 2:1 and 1:1 mixtures of $Fm\bar{3}m$-LuN and $Fm\bar{3}m$-$LuH_3$, respectively. Hence, they appear as alternating layers – with different widths – of octahedral Lu–N bonds and H atoms on tetrahedral and fully occupied octahedral sites. These motifs are not particularly promising in terms of room temperature SC since in high-pressure superhydrides high-$T_c$ SC usually occurs in phases with high H content in which H forms

covalent-metallic bonds either with another H (cage-like hydrides), or with a different element (covalent hydrides).

In addition to metastable, metallic ternary phases, Table 1 and Fig. 2 contain five binary phases (B1-$F\bar{4}3m$-LuH, B2-$Fm\bar{3}m$-LuH, B3-$Fm\bar{3}m$-$LuH_2$, B4-$Fm\bar{3}m$-$LuH_3$, B5-$P6_3/mmc$-$LuH_3$), as well as one additional ternary phase (T1-$P\bar{4}3m$-LuNH). Except for B3-$Fm\bar{3}m$-$LuH_2$, which is situated on the convex hull at 0 GPa, these phases are metastable but have been included nonetheless as they are compatible with the measured XRD spectra and/or have been suggested in other works as viable candidates to explain Dasenbrock-Gammon et al.'s experiments[1]. Also these phases, which are structurally analogous to other low-pressure metal hydrides, such as Pd or Cr hydrides[51,52], are unlikely candidates for room-temperature SC.

Before discussing the superconducting properties in more detail, we briefly compare our calculated phase diagrams with the results of other independent crystal structure searches in the Lu–N–H system, which appeared in the literature during the preparation and revision process of our manuscript[37,53–56]: ref. [53] found six stable binary compounds and no stable ternaries at ambient pressure employing structural templates along with unified input parameters of the ATOMLY materials database. Interestingly, they report a thermodynamically stable $C2/c$-$N_2H_3$ phase, which does not show up in our dataset, even after performing fixed composition calculations on this stoichiometry. The phase diagrams reported by ref. [54], ref. [37], and ref. [56], on the other hand, are consistent with ours, within an accuracy of 5 meV/atom. In particular, ref. [37], who employed the open-source evolutionary algorithm XTALOPT[57], also identified the $P\bar{4}3m$-LuNH phase reported as T1 in Table. 1, and also found it to be highly energetically metastable – i.e., 485 meV/atom (564 meV/atom) above the convex hull according to our (and their) calculations.

**Mining for new superconductors in the Lu–N–H system**

Table 1 summarizes the structural, thermodynamic, and superconducting properties of the structures identified as the most viable candidates for SC. The columns show the unique ID of the structure (B = binary / T = ternary) with a given composition and space group, the calculated distance from the hull ($\Delta E_{hull}$), the logarithmic averaged

**Table 1 | Summary of thermodynamic and superconducting properties for the metallic structures predicted at ambient pressure that may have been found in Dasenbrock-Gammon et al.'s experiment[1]: (i) ternary phases within 50 meV/atom from the convex hull that are also dynamically stable and metallic and (ii) selected binary phases with LuH, LuH$_2$ and LuH$_3$ composition**

| ID | Comp. | SG | $\Delta E_{hull}$ (meV/atom) | $\omega_{log}$ (meV) | $\lambda$ | $\eta$ ($10^4$ meV$^2$) | $T_C^{AD}$ (K) | $T_C^E$ (K) | XRD match | $V_0$ (Å$^3$/$n_{Lu}$) |
|---|---|---|---|---|---|---|---|---|---|---|
| B1 | LuH | $F\bar{4}3m$ | 102 | 14.0 | 0.6 | 0.01 | 3.3 | 3.0 | A | 31.6 |
| B2* | LuH | $Fm\bar{3}m$ | 222 | 24.4 | 0.9 | 0.05 | 17.2 | 18.6 | B | 27.6 |
| B3 | LuH$_2$ | $Fm\bar{3}m$ | 0 | 22.1 | 0.3 | 0.01 | 0.1 | <0.5 | A | 31.6 |
| B4* | LuH$_3$ | $Fm\bar{3}m$ | 101 | 17.6 | 1.7 | 0.05 | 25.4 | 32.0 | A | 31.3 |
| B5* | LuH$_3$ | $P6_3/mmc$ | 10 | 21.5 | 1.7 | 0.08 | 31.9 | 38.5 | none | 34.7 |
| B2 (5 GPa) | LuH | $Fm\bar{3}m$ | – | 24.3 | 0.7 | 0.04 | 9.8 | 9.7 | – | |
| T1 | LuNH | $P\bar{4}3m$ | 485 | 18.5 | 1.0 | 0.03 | 15.6 | 16.0 | A | 31.1 |
| T2 | Lu$_2$NH$_2$ | $P\bar{3}m1$ | 5 | 27.3 | 0.3 | 0.02 | 0.0 | <0.5 | none | 28.9 |
| T3 | Lu$_2$NH$_3$ | $P\bar{3}m1$ | 20 | 38.7 | 0.6 | 0.09 | 10.3 | 10.5 | none | 28.7 |
| T4 | Lu$_3$NH$_4$ | $R\bar{3}m$ | 6 | 26.2 | 0.3 | 0.02 | 0.1 | <0.5 | none | 29.6 |
| T5 | Lu$_3$N$_2$H$_2$ | $R\bar{3}m$ | 2 | 27.2 | 0.2 | 0.02 | 0.0 | <0.5 | none | 28.2 |
| T6 | Lu$_3$N$_2$H$_3$ | $R\bar{3}m$ | 14 | 40.7 | 0.5 | 0.08 | 4.4 | 4.9 | none | 28.0 |
| T7 | Lu$_4$NH$_6$ | $R\bar{3}m$ | 6 | 24.9 | 0.3 | 0.02 | 0.1 | <0.5 | none | 30.1 |
| T8 | Lu$_4$N$_2$H$_5$ | $C2/m$ | 2 | 31.7 | 0.3 | 0.03 | 0.2 | <0.5 | none | 28.7 |
| T9 | Lu$_4$N$_2$H$_5$ | $P2/c$ | 0 | 30.1 | 0.3 | 0.02 | 0.0 | <0.5 | none | 28.7 |
| T10 | Lu$_4$N$_3$H$_2$ | $R\bar{3}m$ | 2 | 29.9 | 0.2 | 0.02 | 0.0 | <0.5 | none | 27.9 |
| T11 | Lu$_5$NH$_8$ | $P\bar{3}m1$ | 6 | 24.9 | 0.3 | 0.02 | 0.1 | <0.5 | none | 30.4 |
| T12 | Lu$_5$N$_4$H$_2$ | $P\bar{3}m1$ | 2 | 30.5 | 0.2 | 0.02 | 0.0 | <0.5 | none | 27.6 |

Asterisks (*) indicate phases which, at the harmonic level, exhibit (few) harmonically-unstable modes, for which el-ph properties were computed integrating only on real (stable) modes. Phases with a Tc > 4 K are discussed in greater detail. In the table, ΔEhull is the energy distance of the compound from the calculated convex; is the logarithmic average phonon frequency; λ is the el-ph coupling strength; η is the McMillan-Hopfield parameter; and are the superconducting critical temperatures estimated from the semi-empirical Allen-Dynes formula and the isotropic Eliashberg equation, respectively; and V0 is the unite cell volume per number of Lu atoms. Corresponding identifiers (ID), compositions (Comp.), space-group (SG), and XRD matches for the phases are provided.

frequency ($\omega_{log}$), and the total *el-ph* coupling parameter ($\lambda$) computed from the Eliasherg functions, as well as the $T_c$ estimated from the semi-empirical Allen-Dynes formula ($T_C^{AD}$)[58] and the isotropic Eliashberg equation ($T_C^E$). In both cases, a standard value of $\mu^* = 0.1$ was assumed for the Morel-Anderson Coulomb pseudopotential. The meaning of the parameter $\eta$ will be discussed in the following. The last two columns of the table indicate whether the calculated XRD pattern matches the experimental pattern reported in ref. 1 for the A (superconducting) or B (non-superconducting) phase, and the last column of the table lists the volume ($V_0$) per number of Lu atoms in the cell, which is a compact indicator of the number of octahedral/tetrahedral site occupation. Phases listed with an asterisk (*) are dynamically unstable at the harmonic level in some part of the Brillouin zone (BZ); the relative *el-ph* properties were obtained integrating over real frequencies.

Most phases listed in Table. 1 have a negligibly small $T_c$ i.e., are not superconducting. This is not surprising, considering that the (Lu+N):H ratio is low (≤3), limiting the possible contribution of hydrogen electronic and vibrational states to SC, and none of the structures examined contain metallic covalent H−H or H−N/Lu bonds.

The most promising ternary structures identified in our search have predicted $T_c$'s lower than 20 K: The $P\bar{4}3m$-LuNH (T1) structure, with $T_c$ = 16.0 K and $\Delta E_{hull}$ = 485 meV/atom; $P\bar{3}m1$-Lu$_2$NH$_3$ (T3), with $T_c$ = 10.5 K and $\Delta E_{hull}$ = 20 meV/atom; and $R\bar{3}m$-Lu$_3$N$_2$H$_3$ (T6), with $T_c$ = 4.9 K and $\Delta E_{hull}$ = 14 meV/atom. A few binary phases, i.e., rocksalt B2-$Fm\bar{3}m$-LuH, B4-$Fm\bar{3}m$-LuH$_3$, and B5-$P6_3/mmc$-LuH$_3$, with $T_c$'s of 18.6, 32.0, and 38.5 K, respectively, even outperform the ternaries.

**Best superconducting candidates**
In the following, we discuss the six phases indicated in Table 1 for which we calculated a non-negligible $T_c$. Figure 3 shows the corresponding phonon dispersions, (partial) phonon density of states (DOS), and isotropic Eliashberg functions $\alpha^2F(\omega)$.

B2-LuH. In our searches, the rocksalt Lu monohydride appears around 222 meV/atom above the convex hull. The H atoms occupy octahedral sites around Lu. With a volume of 27.6 Å, this is one of the densest phases found in our search- In fact, the calculated XRD spectra match almost perfectly the XRD data for the non-superconducting B phase found in ref. 1 experiment. However, in pure B2-$Fm\bar{3}m$-LuH, there should be no Raman-active modes, while ref. 1 Raman spectra exhibit several peaks around 100–250 cm$^{-1}$, probably due to structural distortions, and several peaks at very high frequencies (3000–4000 cm$^{-1}$), indicating trapped H$_2$ or N$_2$ molecules. At ambient pressure, RS-LuH is predicted to be dynamically unstable at the harmonic level – see Fig. 3a. Anharmonic effects are likely to harden the low-lying modes and remove the instability, as in rocksalt PdH[52]. A moderate pressure of 5 GPa has a similar effect, as can be appreciated in Fig. 3a.

Similarly to other low-pressure metal hydrides, the band structure of RS LuH derives from the hybridization of Lu − $d$ and H − $s$ states. Lu-$d$ states fill the gap between H bonding-antibonding states, located at ±7 eV around the Fermi level; the Fermi surface is dominated by Lu-$d$ states, but the small residual H hybridization is sufficient to boost the $T_c$, providing a finite coupling with H modes. Integrating the Eliashberg function only on real frequencies yields $\omega_{log}$ = 24 meV, $\lambda$ = 0.9, providing a $T_c$ of 18.6 K.

B4-LuH$_3$. $Fm\bar{3}m$-LuH$_3$ is the structure originally proposed by ref. 1 for the high-$T_c$ phase A, although later studies have suggested, instead, the non-superconducting B3-$Fm\bar{3}m$-LuH$_2$ phase, which has the same *fcc* Lu sublattice and very close unit cell volume[28,29,31–35]. $Fm\bar{3}m$-LuH$_3$ appears about 100 meV/atom above the convex hull ($P6_3$-LuH$_3$ turns out to be thermodynamically more stable). It crystallizes in the well-known D0$_3$ structure, in which the metal atoms form a face-centered-cubic lattice and hydrogen occupies all the tetrahedral and octahedral interstitial sites. Indeed, the simulated XRD spectrum agrees with the Bragg peaks of the A phase in ref. 1. However, the fully symmetric

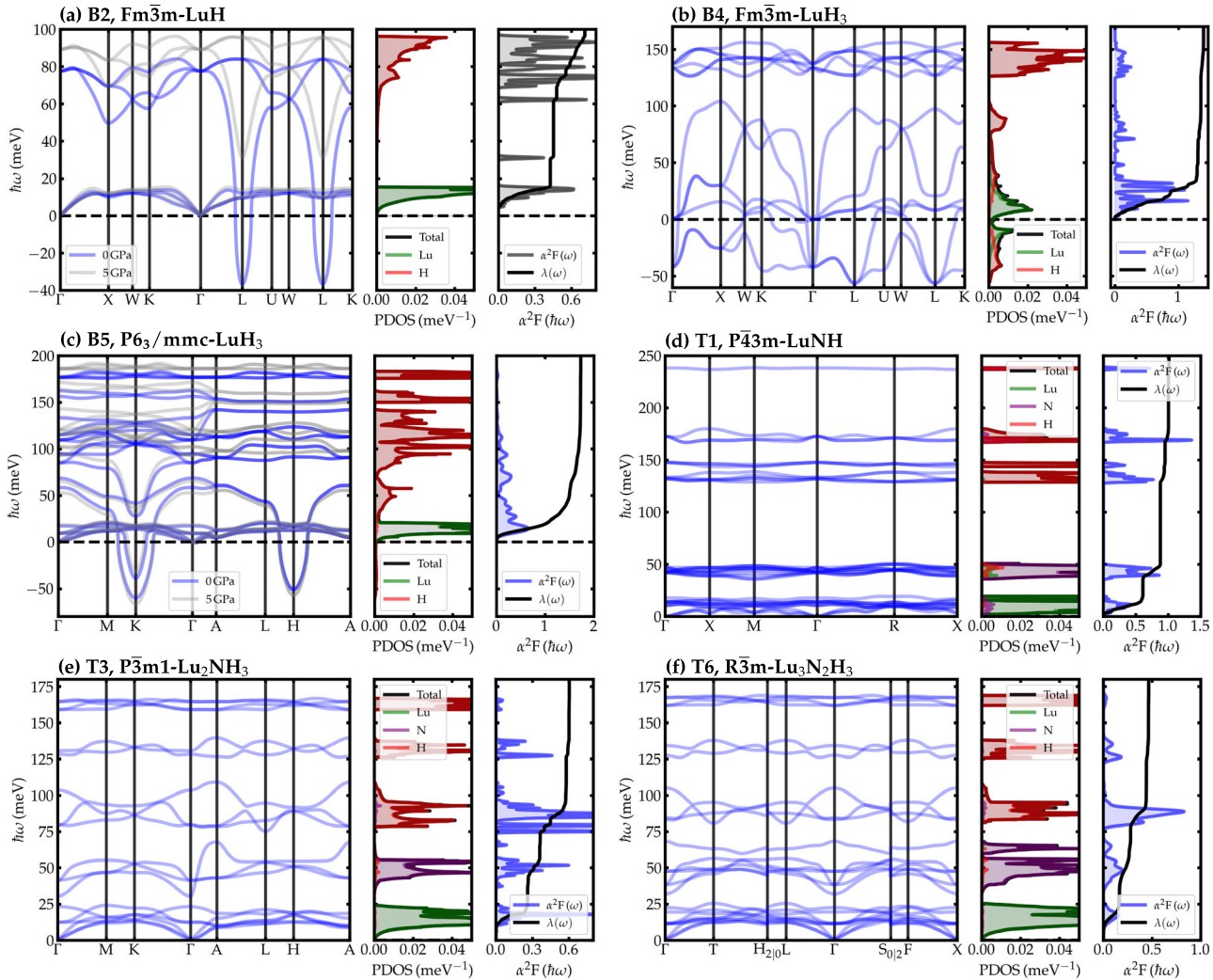

**Fig. 3 | Phononic and superconducting properties of the best superconducting candidates.** Phonon band structure (solid blue and light gray lines), phonon density of states (Lu PDOS in shaded green, N PDOS in shaded purple, and H PDOS in shaded red), isotropic Eliashberg function $\alpha^2F$ (shaded gray), and total *el-ph* coupling parameter $\lambda$ (solid black lines) for (**a**) B2, $Fm\bar{3}m$-LuH, (**b**) B4, $Fm\bar{3}m$-LuH$_3$, (**c**) B5, $P6_3/mmc$-LuH$_3$, (**d**) T1, $P\bar{4}3m$-LuNH, (**e**) T3, $P\bar{3}m1$-Lu$_2$NH$_3$, and (**f**) T6, $R\bar{3}m$-Lu$_3$N$_2$H$_3$.

$Fm\bar{3}m$ phase should possess a single Raman active mode, for which we computed a frequency of 1079 cm$^{-1}$. Dasenbrock-Gammon et al.'s spectra comprise at least 11 peaks, with frequencies ranging from 100 to 1220 cm$^{-1}$.

As can be appreciated in Supplementary Fig. 9, the electronic structure comprises a metallic state with 3 bands crossing $E_F$. The Fermi level is right below a steep shoulder in the DOS, formed by unoccupied Lu-d bands, while occupied states are of mixed Lu–H character.

Our calculations show that at the harmonic level, several unstable modes exist in the entire BZ - Fig. 3b. Calculations by other authors show that the dynamic instability is not removed by moderate pressure and/or N substitution[37,54].

To obtain an estimate of $T_c$, we integrate the Eliashberg function only on real frequencies, obtaining $\lambda = 1.65$ and $\omega_{log} = 17.6$ meV, resulting in $T_c = 25.4$ K; the $T_c$ obtained by solving the isotropic Eliashberg equations is higher ($T_c = 32$ K), but still one order of magnitude too low for ambient SC.

Some of us have recently shown that $Fm\bar{3}m$-LuH$_3$ could be stabilized by quantum anharmonic lattice effects at near ambient pressures for temperatures above 200 K[49]. Increasing the pressure up to 6 GPa the temperature required for stability is reduced to $T > 80$ K.

Still, the $T_c$ for the quantum anharmonic- and temperature-stabilized $Fm\bar{3}m$-LuH$_3$ phase is predicted to be between 50 and 60 K, i.e., well below RT, and in fact even well below the temperatures required to dynamically stabilize the structure[49].

B5-LuH$_3$. $P6_3/mmc$-LuH$_3$ is only 10 meV/atom above the hull and assumes the Na$_3$As-prototype structure, which comprises two inequivalent Lu sites, with the first 4-coordinated to four equivalent H atoms and the second bonded in a trigonal planar geometry to three equivalent H atoms. As expected, the hexagonal symmetry does not produce any sizable match with the experimental XRD pattern of the A or B phases.

The Fermi level lies in a pseudogap formed by two-dimensional H-s hole-pockets around the zone center and Lu-d electron-pockets around the BZ corners, giving rise to a compensated (equal amounts of holes and electrons), low DOS at $E_F$.

As shown in Fig. 3c, the phonon dispersion reveals unstable modes around the $K$ and $H$ high-symmetry points in BZ, again indicative of lattice instabilities. We have checked that pressures up to 5 GPa cannot suppress the dynamic instability.

To estimate $T_c$, we again set the *el-ph* matrix elements of all imaginary modes to zero. As a result, we get a high *el-ph* coupling constant $\lambda = 1.7$ and $\omega_{log} = 21.5$ meV, resulting in $T_c = 31.9$ K (38.5 K) from

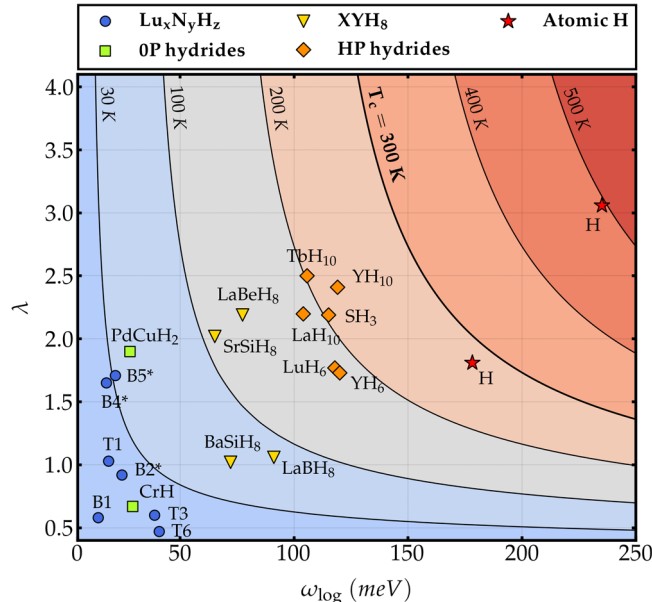

**Fig. 4 | Electron–phonon coupling strength $\lambda$ as a function of the logarithmic average phonon frequency $\omega_{\log}$ for different classes of superconducting hydrides.** The best Lu--N--H hydrides considered in this work are indicated by blue circles and a selection of other hydrides is included as reference. Contour lines for $T_c$ are plotted according to Eq. (1) with $\mu^* = 0.1$.

McMillan-Allen-Dynes formula (isotropic Eliashberg equations). Again, this value is too low to be compatible with room-temperature SC, even if the effects of anharmonicity or impurities are taken into account.

T1-LuNH. T1-LuNH is found 485 meV/atom above the convex hull in our search. It assumes a rocksalt structure with a *fcc* Lu-Lu sublattice with $a = 4.990$ Å, which matches considerably well the collected Bragg peak positions for the A phase (by comparing the simulated and measured XRD diffractograms, we obtain a normalized mean squared error of 0.17), representing 92.25% of the measured sample[1].

The structure is dynamically stable in the whole BZ at the harmonic level, but the calculated frequencies for the Raman-active modes do not match the main peaks of sample A reported in ref. 1, indicating again a strong effect of anharmonicities in the calculations and/or impurities or disorder in the measured samples.

LuNH is a multi-band metal with four bands crossing the Fermi level, which provide a high DOS at $E_F$ of 2.57 states/eV. Most of these states, however, have Lu-*d* orbital character, representing 70% of the total DOS at $E_F$ (see Supplementary Fig. 9); The H states only constitute 3.8% of the states around the Fermi level and thus do not considerably contribute to any *el-ph* coupling.

The total *el-ph* coupling $\lambda = 1.0$ is sizeable, but, as can be appreciated in Fig. 3d, originate mainly from low-frequency Lu and N vibrations. Hence, the corresponding $\omega_{\log}$ is very small (19 meV), and thus $T_c$ is predicted to be about 16 K.

T3-$Lu_2NH_3$. $Lu_2NH_3$ is 20 meV/atom away from the convex hull at 0 GPa but is on the hull at 10 GPa, as shown in Supplementary Fig. 4. Its crystal structure resembles the well-known $CdI_2$ type, with one H atom being six-fold coordinated to Lu atoms to form edge-sharing $LuH_6$ octahedra. Nitrogen is intercalated between the lutetium layers, and the two additional H atoms are situated at the 2*d* Wyckoff position interstices between adjacent $LuH_6$ octahedra.

We find that the simulated XRD patterns match neither of the two patterns reported by ref. 1.

Interestingly, as shown in Supplementary Fig. 9, the electronic structure of $Lu_2NH_3$ shares some similarities with transition metal dichalcogenides[59] and $MgB_2$[60]. The Fermi surface topology resembles

that of the prototypical charge-density-wave superconductors $TiSe_2$[61], and $ZrTe_2$[62]: Two bands are crossing the Fermi level: a hole-like band crosses $E_F$ around the Γ-A high-symmetry line, giving rise to out-of-plane quasi-cylindrical pockets derived from H-*s* orbitals, whereas an electron-like band derived from the Lu-*d* manifold crosses the Fermi level around the *L* direction in BZ. The strongly nested Fermi surface causes a largely anisotropic distribution of the *el-ph* coupling.

As can be appreciated in Fig. 3e, the vibrational frequencies extend up to 175 meV, and most of the *el-ph* coupling originates from the H-dominated phonon modes with energies between 75 and 110 meV. However, both the N-dominated modes (30–55 meV) as well as the Lu-dominated modes (below 25 meV) also contribute to $\lambda$. In contrast, the high-frequency vibrations of H (above 110 meV) give no significant *el-ph* contribution. In the end, we find $\lambda = 0.5$ and $\omega_{\log} = 41$ meV, resulting in a $T_c = 5$ K.

From the electronic dispersions and the momentum distribution of electron–phonon interactions of $Lu_3NH_3$, and their similarities with $ZrTe_2$, $TiSe_2$, and $MgB_2$, one can derive some intriguing conclusions that indicate that $Lu_3NH_3$, if stabilized, could be a promising platform to observe several exotic effects. First, the disconnected two-band Fermi surface with strongly distinct orbital projections could give rise to a two-gap superconducting state, which could significantly enhance the $T_c$[63]. Multi-gap SC may also give rise to an unusual response of the superconducting state to magnetic fields[64,65]. Second, the low $N(E_F)$ of 0.25 states/eV and compensated electronic character, with nearly equal electron- and hole-type carriers at the Fermi surface, can cause the formation of electron-hole bound states[66], suggesting that excitonic contributions could be crucial in this system. Finally, the presence of hole-type H-*s* cylinders at the center of BZ leaves room for optimizing the $T_c$ by increasing the hole pocket size through charge or chemical doping and strain engineering.

T6-$Lu_3N_2H_3$. $R\bar{3}m$-$Lu_3N_2H_3$, which is only 14 meV/atom away from the convex hull, has the trigonal crystal structure of delafossite with H atoms in tetrahedral and octahedral sites, as schematically shown in Fig. 2. Again, we find no match with the experimental XRD patterns. Similarly to the previously discussed $P\bar{3}m1$-$Lu_2NH_3$, the Fermi level is almost at the bottom of a pseudogap, resulting in a relatively low $N(E_F)$.

The largest contributions to $\lambda$ originate from low-frequency Lu vibrations and the lowest branch of H vibrations (75–100 meV), as indicated in Fig. 3f. Thus, both $\lambda$ and $\omega_{\log}$ are comparatively low (0.5 and 41 meV, respectively), resulting in a $T_c$ of about 5 K.

## How likely is conventional room-temperature superconductivity in Lu−N−H?

None of the metallic phases identified through our high-throughput screening of the Lu−N−H ternary hull at ambient pressure is compatible with the report of room-temperature SC by ref. 1. Indeed, the calculated $T_c$'s are below those predicted for many ambient-pressure metal hydrides and well below those anticipated for ternary sodalite-clathrate structures of La-B/Be or Ba/Sr-Si hydrides ($T_c \lesssim 120$ K), which, according to calculations and recent experiments, may be quenched to near-ambient pressure[16–18,67,68].

Figure 4 compares the *el-ph* properties of our best Lu−H−N superconductors (blue circles) with those of other families of H-based superconductors, i.e., metal hydrides at ambient pressure (green squares), ternary $XYH_8$ sodalite-clathrate hydrides (yellow triangles), high-pressure binary hydrides (orange diamonds), and metallic hydrogen (red stars), on a $\omega_{\log}$–$\lambda$ diagram. The isocontours and color scale are obtained using the McMillan-Allen-Dynes formula[58] with $\mu^* = 0.1$:

$$k_B T_c^{AD} = \frac{\omega_{\log}}{1.2} \exp\left[-\frac{1.04(1+\lambda)}{\lambda(1-0.62\mu^*)-\mu^*}\right]. \quad (1)$$

Data were collected from refs. [5,13,18,51,67,69-73] — additional details can be found in Supplementary Information.

Compounds belonging to different families cluster around different $T_c$ isolines in ascending synthesis pressure. The increase in $T_c$ seems to be solely driven by an increase of the logarithmically-averaged phonon frequency $\omega_{\log}$, which varies from ~30 meV in ambient-pressure metal hydrides to 230 meV in atomic hydrogen at 2 TPa. $\omega_{\log}$ essentially measures the average stiffness of the phonon modes involved in the superconducting pairing.

A second material-dependent parameter, $\eta$, can be introduced to quantify the intensity with which these modes couple to electrons. $\eta$ is related to the total *el-ph* coupling $\lambda$ through the McMillan-Hopfield's formula $\lambda = \eta/\omega^2$ [74], where $\omega$ is an average phonon frequency.

We have computed the values of $\eta$ for all compounds in Fig. 4. Values of $\eta$ range from $\eta = 10^2$ meV$^2$ in Lu−N−H and other low-P binary hydrides, to $10^4$ meV$^2$ in binary and ternary sodalite-clathrate hydrides, and up to $10^5$ meV$^2$ in atomic hydrogen − details in Supplementary Information. Differences in $\eta$ typically reflect differences in chemical bonding, with larger values indicating more localized, directional bonds [13,75].

Our analysis leads us to two main conclusions regarding Dasenbrock-Gammon et al. data [1]: (i) All superconducting Lu−N−H phases identified in this work are essentially homogeneous and closely related to other low-pressure metal hydrides: SC is dominated by the metal sublattice, and hydrogen only plays a minor role, providing a marginal boost to the metal's $T_c$ [13]. Given that both $\omega_{\log}$ and $\eta$ are one to two orders of magnitude too low for room-temperature SC, it is implausible that any renormalization effects due to distortions, impurities, or phonon anharmonicities may be invoked to explain Dasenbrock-Gammon et al. data [1]; (ii) The only possibility to explain room-temperature SC is to hypothesize that an exotic phase has been realized in experiments where an extremely large *el-ph* coupling is concentrated in a small fraction of high-frequency modes ($\omega_{\log} > 150$ meV), like those provided by localized H−H or N−H vibrations. In fact, despite the dense hydrogen sublattice, even cage-like ternary sodalite clathrate hydrides cannot support $T_c$'s higher than 120 K at ambient pressure due to too low $\omega_{\log}$ and $\eta$. A preliminary scan of ternary structures in a reasonable metastability range can rule out this possibility.

Two recent works that appeared during the revision of our work [76,77] seem to suggest that the inclusion of strong correlation effects in the Lu-N-H system by the so-called LDA+U method [78] may strongly affect our conclusions since the addition of a finite U on Lu leads to a rearrangement of the electronic structure, which in some particular structural models can sensibly boost the value of the DOS at the Fermi level.

However, it is well known that: (i) the relative band positions in the LDA+U approximation depend sensibly on the chosen value of $U$; (ii) values of $U$ computed even in the most accurate constrained-random-phase approximation for the same compound may fluctuate depending on the details of the projection/downfolding procedure [79]; in fact, ref. 80 have shown, by hybrid functional calculations, that different phases require different Hubbard potentials to describe the $f$ electrons correctly. Even more crucial for the conclusions of the present work, however, is the fact that: (iii) including finite-bandwidth corrections in the Eliashberg theory for conventional SC washes out the effect of sharp peaks in the DOS [81,82] and (iv) as shown in Fig. 4, even tripling the *el-ph* coupling constant of any of the predicted Lu-N-H phases would not be sufficient to bring the system even close to room-temperature SC without a simultaneous increase of the phonon frequency by an order of magnitude.

In summary, we have investigated the phase diagram and SC of the Lu−N−H system using state-of-the-art methods for crystal structure prediction.

The phase diagram of the Lu−N−H system is essentially determined by the thermodynamically stable binary phases $Fm\bar{3}m$-LuN and $Fm\bar{3}m$-LuH$_2$, and, to a lesser extent, $P2_13$-NH$_3$. As a result, the ternary phases closest to the hull can be described as layered mixtures of the *fcc* LuN and LuH$_2$ phases or as H atoms trapped in an *fcc* LuN lattice. Some of these phases do match the XRD patterns reported by ref. 1, suggesting that the diffusion of N or H into a rock-salt LuN sublattice, which is justified from a thermodynamical point of view, may explain the formation of ternary phases in experiments. We note, however, that none of the stoichiometrically pure phases described here matches the reported Raman spectra, indicating a likely presence of impurities or distortions in the experimental samples.

A direct calculation of the superconducting properties of all metastable predicts no superconductor with a $T_c$ higher than 40 K, almost an order of magnitude less than the result of ref. 1. Indeed, electronic structure calculations do not support high-$T_c$ conventional SC in any of the examined structures.

Our results demonstrate unambiguously that Lu-hydrides, whether doped with N or not, cannot harbor ambient SC within the *el-ph* mechanism. The high fraction of Lu-$d$ states at the Fermi level, the weak coupling between the already scarce low-energy H-$s$ states with the high-frequency optical modes, and the substantially low $\omega_{\log}$ make the Lu-H-N system highly unlikely to meet the extraordinary conditions required for ambient SC.

While we acknowledge that strong electronic correlations may have a quantitative impact on the electronic properties of some of the calculated phases, the discrepancy between our calculated phonon frequencies and electron−phonon matrix elements and the measured $T_c$ is too large to be explained by any renormalization effect on the electronic DOS [76,77]. Therefore, we maintain that unless experiments unambiguously demonstrate the synthesis of such an exotic phase, the century-old *Sisyphus'* quest of ambient SC still remains open.

## Methods

### Crystal structure predictions

Crystal structure predictions were carried out using two different methods: (i) evolutionary algorithms as implemented in the USPEX package [42,43] and (ii) ab initio random structure searching (AIRSS) [39,40]. The structures resulting from the two runs were then relaxed using the same numerical parameters and merged into a single convex hull. In the following, we summarize the technical details of the USPEX search, the AIRSS search, and the final relaxation. Further computational details are provided in Supplementary Methods.

USPEX. We performed two sets of ternary and binary variable-composition searches at 0 and 10 GPa employing cells with 8−16 and 12−24 atoms. Pseudo-binary searches were also conducted along the selected lines corresponding to all possible two-phase reactions between the known stable binaries. Additionally, we also performed fixed-composition structural searches along the LuNH$_x$ ($x = 1, 2, ..., 23$) stoichiometries. Each structure was fully relaxed using VASP [83] employing a 5-step relaxation procedure. In total, more than 100,000 structures were sampled in our USPEX searches.

AIRSS. We used AIRSS accelerated by ephemeral data-derived potentials [41] (EDDPs, the details of which are presented in the Supplementary Information) to search for structures in the Lu−N−H and LuH$_2$−LuN−LuH$_3$ ternary and pseudo-binary systems at 0, 2 and 10 GPa. Approximately 100,000 structures were calculated in this way. The speed of the potential enables large unit cells, containing up to 64 atoms, to be sampled in this case. The best structures from the EDDP calculations were then used for subsequent DFT calculations using CASTEP [84]. We used the on-the-fly generated ultrasoft, 'QC5' pseudo-potentials, a plane-wave cut off of 440 eV, and k-point spacing of $0.05 \times \frac{2\pi}{\text{Å}}$ for searching and training the potentials.

Convex hull. To generate the convex hull of the combined AIRSS and USPEX data, we performed geometry optimizations of all

structures using the same parameters as in the AIRSS searches. All structures within 50 meV of the hull at 0, 2 or 10 GPa were retained for final, well-converged calculations using 'C19' pseudopotentials with a plane-wave cut-off of 1000 eV, and k-point spacing of $0.02 \times \frac{2\pi}{\text{Å}}$.

## Electronic and vibrational properties

Electronic and vibrational properties were computed using the QUANTUM ESPRESSO[85,86] suite, using scalar-relativistic optimized norm-conserving Vanderbilt pseudopotentials (ONCV)[87,88] and a PBE-GGA parametrization for the exchange and correlation functional[89]. Kohn-Sham orbitals were expanded on plane-waves, with a kinetic energy cutoff of 100 Ry for the wavefunctions and $8 \times 8 \times 8$ unshifted **k**-grid sampling over the BZ[90] with a Methfessel-Paxton gaussian smearing[91] of 0.04 Ry. Phonon frequencies were obtained by Fourier interpolation of the dynamical matrices on $2 \times 2 \times 2$ **q**-grid within Density Functional Perturbation Theory (DFPT)[92]. Electron–phonon properties were computed on $16 \times 16 \times 16$ **k**-grid. For structures with $T_c$ higher than 10 K, we re-computed the dynamical matrices on a $6 \times 6 \times 6$ **q**-grid and the *el-ph* properties on $30 \times 30 \times 30$ **k**-grid.

## Data availability

The data generated in this study have been deposited in the Zenodo database under accession code 7839254[93] and in the Supplementary Information file.

## Code availability

The codes generated in this study have been deposited in the Zenodo database under accession code 7839254[93].

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

## Acknowledgements

We would like to acknowledge Roman Lucrezi, Angela Rittsteuer, and Markus Aichhorn for insightful discussions. PPF and LTFE gratefully acknowledge the São Paulo Research Foundation (FAPESP) under Grants 2020/08258-0 and 2021/13441-1. CH acknowledges the Austrian Science Fund (FWF): P 32144-N36. LB acknowledges support from Project PE0000021,"Network 4 Energy Sustainable Transition - NEST", funded by the European Union - NextGenerationEU, under the National Recovery and Resilience Plan (NRRP), Mission 4 Component 2 Investment 1.3 - Call for tender No. 1561 of 11.10.2022 of Ministero dell'Universitá e della Ricerca (MUR). Calculations were performed on the Vienna Scientific Cluster (proj. 71754 "TEST"), on the dcluster of the Graz University of Technology, on the CINECA cluster (proj. IsC99-ACME-C); Computational resources for the UK partners were provided by the UK's National Supercomputer Service through the UK Car-Parrinello consortium (EP/P022561/1) and the Cambridge Service for Data Driven Discovery (CSD3) using Tier-2 EPSRC funding (EP/T022159/1).

## Author contributions

P.P.F. and L.J.C. contributed equally. P.P.F., L.J.C., A.C., C.H. and L.B. wrote the main draft. P.P.F. performed the USPEX ternary, pseudo-binaries, and fixed composition searches. E.K. performed the USPEX binary searches. C.J.P. and L.J.C. carried out the AIRSS searches and made the final relaxation. P.P.F. and A.C. carried out the phonon and *el-ph* calculations. A.C. and L.B. developed the effective superconducting model. L.J.C., S.D.C. and F.G. conducted the HT screening. F.G. performed the XRD pattern assessment and C.H. the Raman analysis. P.P.F., A.C., S.D.C. and L.T.F.E. prepared the figures and tables. C.J.P., C.H. and L.B. supervised this project. All authors participated in the discussions and revised the manuscript.

## Competing interests

The authors declare no competing interests.
