## [Peer Review File · Nature Communications]

Search for ambient superconductivity in the Lu-N-H systemREVIEWER COMMENTS

Reviewer #1 (Remarks to the Author):

Comments:

Pedro P. Ferreira et al performed a comprehensive, detailed study of the phase diagram of the Lu–N–H system, looking for superconducting phases. They identified 52 viable candidates for conventional superconductivity, but none were compatible with the room-temperature superconductivity reported by Dasenbrock et al, which confirms that no conventional mechanism can explain the reported high T_c in Lu–N–H. The results of this works are attractive. Therefore, it is recommended to be published after made the following revisions.

1. On page 3, the sentence “two metastable binaries (B3-Fm $\bar{3}$ m-LuH₂, B5-P63/mmc-LuH₃)” is inaccurate. B3-Fm $\bar{3}$ m-LuH₂ is on the convex hull at 0 GPa and in the caption of Fig. 1, blue circles indicate the thermodynamically stable phases. By the way, we did not see where B5-P63/mmc-LuH₃ is in Fig. 1, are P63/mmc-LuH₃ and P63-LuH₃ the same structure?

2. The pseudopotential of lutetium should be tested and compared with the known experimental results for example LuN, LuH, LuH₂, LuH₃. In addition, in general, d or f electrons have strong correlation. Did the authors consider DFT+U?

3. Crystal structures of the best candidates for SC in Lu–N–H ternary system are shown in Figure 2. In T3, T6, T8, there are some atomic H in the structure, is that true? ELF's should be given.

4. Some calculated critical temperatures seems much smaller than the values previously reported in Table IV of SM. Any explanation/justification? The authors should carefully check the data given in the table.

5. There are some minor English expression errors (typos and format). For example:

- “NH₃–Lu₃”
- The table title should be written at the top of the Table IV in SM.

- The format of the space group contained in Figure.3 on page 6 needs to be modified. In addition, similar ones in SM also need to be changed.

6.The page numbers for many references are missing, for example, 8, 9, 19, 22, 23 and so on. By the way, is reference 11 still in press? Please check it.

7.Did the authors test the effect of different q-grids and k-grids? If so, please list the tested results

8.Is the el-ph calculations still reasonable with so large and many imaginary frequencies for B4-LuH3?

Reviewer #2 (Remarks to the Author):

Recent report of room temperature superconductivity in a lutetium hydride at 1 GPa took the scientific community by surprise and excited unprecedented activity by other groups, all of which hurried to prove that report wrong. I agree with that conclusion, but in my opinion, more important than proving an obviously wrong paper, is to learn new insights about where to look for new superconductors and where not.

Present paper gives a compelling computational proof that LuH₂-LuH₃ cannot be a room T_c superconductor and explains why.

I recommend publishing this work.

Artem R. Oganov

Reviewer #3 (Remarks to the Author):

Hydrides have become the most promising candidates for room-temperature superconductivity. However, their superconducting transformation pressures are extremely high (usually above 200 GPa), which is unfavorable for real-world applications. Therefore, achieving the superconductivity of hydrides under low pressure has become the focus of academic attention. Recently, Dasenbrock et al. reported that N-doped lutetium hydride has shown a room-temperature superconductivity at near-ambient pressure. Then, a large number of theoretical and experimental scientists are attempting to confirm this discovery. In this work, the authors have explored over 200,000 candidate structures. None of them has room-temperature superconductivity. This study precludes the conventional superconducting mechanism of Lu-N-H. Overall, the adopted theoretical methods are appropriate, and the calculated results are reliable and significant for the research of hydrides. But there are some issues needed to be addressed.

1. The hydrogen content in hydrides has a great influence on superconductivity. The author should justify the rationality of selecting stoichiometry.
2. There have been many theoretical calculations supporting Lu-N-H system with the lower superconducting transition temperatures, but their research perspectives and methods are different. It is recommended that the authors summarize these studies appropriately.
3. Could the authors discuss more the similarities between Lu₂NH₃, TMD, and MgB₂?

Reviewer 1

Pedro P. Ferreira et al performed a comprehensive, detailed study of the phase diagram of the Lu–N–H system, looking for superconducting phases. They identified 52 viable candidates for conventional superconductivity, but none were compatible with the room-temperature superconductivity reported by Dasenbrock et al, which confirms that no conventional mechanism can explain the reported high T_c in Lu–N–H. The results of this works are attractive. Therefore, it is recommended to be published after made the following revisions.

1. **On page 3, the sentence “two metastable binaries (B3- $Fm\bar{3}m$ -LuH₂, B5- $P63/mmc$ -LuH₃)” is inaccurate. B3- $Fm\bar{3}m$ -LuH₂ is on the convex hull at 0 GPa and in the caption of Fig. 1, blue circles indicate the thermodynamically stable phases. By the way, we did not see where B5- $P63/mmc$ -LuH₃ is in Fig. 1, are $P63/mmc$ -LuH₃ and $P63$ -LuH₃ the same structure?**

A: We rephrased the “two metastable binaries” sentence to “In addition to metastable, metallic ternary phases, Tab.1 and Fig.2 contain five binary phases (B1- $F\bar{4}3m$ -LuH, B2- $Fm\bar{3}m$ -LuH, B3- $Fm\bar{3}m$ -LuH₂, B4- $Fm\bar{3}m$ -LuH₃, B5- $P63/mmc$ -LuH₃), as well as one additional ternary phase (T1- $P\bar{4}3m$ -LuNH), which have been included because, although highly metastable (except for B3- $Fm\bar{3}m$ -LuH₂, which is situated at the convex hull at 0 GPa), they are compatible with the measured XRD spectra and have been suggested by different authors as viable candidates to explain Dasenbrock-Gammon *et al.*'s experiments [1]”.

On the LuH₃ phases, $P63/mmc$ -LuH₃ and $P63$ -LuH₃ are indeed different phases with different crystallographic space groups and formation energies. We don't see B5- $P63/mmc$ -LuH₃ in Fig. 1 because this phase is situated 10 meV/atom above the convex hull, while $P63$ -LuH₃ is the one located on the convex hull.

2. **The pseudopotential of lutetium should be tested and compared with the known experimental results for example LuN, LuH, LuH₂, LuH₃. In addition, in general, d or f electrons have strong correlation. Did the authors consider DFT+U?**

A: We have used the SG15 Optimized Norm-Conserving Vanderbilt (ONCV) pseudopotentials [2, 3], which are widely tested and applied successfully in many different applications. Our band structures, phonons, and electron-phonon properties obtained by using the SG15 database are all consistent with the follow-up works released concomitantly/after ours, which, in these cases, employ different codes and pseudos.

We have also not considered the DFT+U approximation in our calculations. Indeed, we touched upon the issue of describing the *f*-states of Lu in the last paragraph of the *Methods* section. In all compounds investigated, the 4*f* orbitals give rise to highly localized, non-dispersive bands below -4 eV with virtually zero contribution in the close vicinity of the Fermi level, which indicates a null charge transfer from the 4*f* manifold to the H-*s*, N-*p*, or

Lu-*d* shells. This is an essential indication that the on-site Coulomb interactions of the *f* electrons do not play any crucial role in SC or stability. As highlighted by the Referee, several follow-up studies investigating strong correlation effects on LuH₂, and LuH₃ have appeared during the revision phase of our manuscript [1, 4, 5]. We agree that these results deserve to be discussed in our manuscript.

Therefore, we have added the following paragraph to the discussion of the results: "*However, it is well known that: i) the relative band positions in the LDA+U approximation depend sensibly on the chosen value of U; ii) values of U computed even in the most accurate constrained-random-phase approximation for the same compound may fluctuate depending on the details of the projection/downfolding procedure [6]; in fact, Wu et al. [5] have shown, by hybrid functional calculations, that different phases require different Hubbard potentials to describe the f electrons correctly. Even more crucial for the conclusions of the present work, however, is the fact that: iii) including finite-bandwidth corrections in the Eliashberg theory for conventional superconductivity washes out the effect of sharp peaks in the DOS [7, 8] and iv) as shown in Fig. 4, even tripling the el-ph coupling constant of any of the predicted Lu-N-H phases would not be sufficient to bring the system even close to room-temperature superconductivity without a simultaneous increase of the phonon frequency by an order of magnitude.*"

3. **Crystal structures of the best candidates for SC in Lu–N–H ternary system are shown in Figure 2. In T3, T6, T8, there are some atomic H in the structure, is that true? ELF's should be given.**

A: There is no atomic H in T3, T6, and T8. The H-H first-neighbor distance is 2.13 Å for T3, 2.11 Å for T6, and 2.18 Å for T8. The ELF's for the structures shown in Figure 2 of the main manuscript are now shown in the SM, as asked by the referee.

4. **Some calculated critical temperatures seems much smaller than the values previously reported in Table IV of SM. Any explanation/justification? The authors should carefully check the data given in the table.**

A: To make the comparison with our data consistent, the values reported in Table IV of SM were recomputed by employing the McMillan-Allen-Dynes formula with $\mu^* = 0.1$ using the ω_{\log} and λ collected in the literature, as cited in Table IV. However, as well known, the McMillan-Allen-Dynes formula underestimates the critical temperature compared to the full solution of the Eliashberg equations. We rephrased the caption of Table IV to make this point clearer. The caption now reads: "*Superconducting properties of selected hydrides from literature. To make the comparison with our data consistent, the superconducting critical temperatures (T_c^{AD}) reported in this table were re-computed by employing the semi-empirical McMillan-Allen-Dynes formula with $\mu^* = 0.1$ using the ω_{\log} and λ collected in the literature, as cited in the "Ref" column.*"

5. **There are some minor English expression errors (typos and format). For example: (i) NH3–Lu3; (ii) The table title should be written at the top of the Table IV in SM; (iii) The format of the space group contained in Figure.3 on page 6 needs to be modified. In addition, similar ones in SM also need to be changed.**

A: We thank the Referee for such careful reading. We corrected all typos and format issues pointed out above and also made a careful, thorough reading through the whole text, searching for additional text/formatting issues.

6. **The page numbers for many references are missing, for example, 8, 9, 19, 22, 23 and so on. By the way, is reference 11 still in press? Please check it.**

A: We thank the Referee for pointing this out. We revised all references and corrected all issues raised above.

7. **Did the authors test the effect of different q-grids and k-grids? If so, please list the tested results**

A: We have employed a $8 \times 8 \times 8$ k-grid sampling over the BZ for all calculations to compute the DFT charge density. Electron-phonon properties were computed on $16 \times 16 \times 16$ k-grid and $2 \times 2 \times 2$ q-grid on a high-throughput level. For structures with T_c higher than 10 K, we re-computed the dynamical matrices on a $6 \times 6 \times 6$ q-grid and the *el-ph* properties on $30 \times 30 \times 30$ k-grid.

In another recent work [Saha *et al.*, Phys. Rev. Materials 7, 054806 (2023)] some of us have compiled a database of over more than 100 superhydrides, for which we have performed extensive convergence tests.

Based on this experience and in agreement with other works in the literature on Lu-H-N (see, for example, Hilleke et al. [9] and Huo et al. [10]), the chosen k- and q-grids are enough to guarantee a convergence lower than 10 meV/atom.

Moreover, we included the results of convergence tests for LuH₃ in the SM.

8. Is the el-ph calculations still reasonable with so large and many imaginary frequencies for B4-LuH₃?

A: The T_c of B4-LuH₃ was obtained by integrating the Eliashberg function only on real frequencies, thus obtaining $\lambda = 1.65$ and $\omega_{\log} = 17.6$ meV, resulting in $T_c = 25.4$ K, as reported in the main manuscript. Therefore, this is a rough estimation and serves as indication of the tendency of the material to conventional SC. Some of the present authors performed a further study using more elaborate methods to evaluate quantum anharmonic effects on the stability and T_c of the B4-LuH₃ phase – Ref. [49]. In that work it is demonstrated that $Fm\bar{3}m$ -LuH₃ is stabilized at near ambient pressures for temperatures above 200 K when including quantum anharmonic lattice effects. By increasing the pressure up to 6 GPa, the required temperature for stability is reduced to $T > 80$ K. However, even including quantum lattice effects the calculated T_c remains well below room temperature ($T_c = 50 - 60$ K).

Reviewer 2

Recent report of room temperature superconductivity in a lutetium hydride at 1 GPa took the scientific community by surprise and excited unprecedented activity by other groups, all of which hurried to prove that report wrong. I agree with that conclusion, but in my opinion, more important than proving an obviously wrong paper, is to learn new insights about where to look for new superconductors and where not. Present paper gives a compelling computational proof that LuH₂-LuH₃ cannot be a room T_c superconductor and explains why.

I recommend publishing this work.

Artem R. Oganov

A: We thank Prof. Artem R. Oganov for such careful reading and for expressing appreciation of our work.

Reviewer 3

Hydrides have become the most promising candidates for room-temperature superconductivity. However, their superconducting transformation pressures are extremely high (usually above 200 GPa), which is unfavorable for real-world applications. Therefore, achieving the superconductivity of hydrides under low pressure has become the focus of academic attention. Recently, Dasenbrock et al. reported that N-doped lutetium hydride has shown a room-temperature superconductivity at near-ambient pressure. Then, a large number of theoretical and experimental scientists are attempting to confirm this discovery. In this work, the authors have explored over 200,000 candidate structures. None of them has room-temperature superconductivity. This study precludes the conventional superconducting mechanism of Lu-N-H. Overall, the adopted theoretical methods are appropriate, and the calculated results are reliable and significant for the research of hydrides. But there are some issues needed to be addressed.

1. The hydrogen content in hydrides has a great influence on superconductivity. The author should justify the rationality of selecting stoichiometry.

A: As discussed in the main manuscript, we have performed a high-throughput screening on Lu-N-H ternary system. Out of all stoichiometries sampled, we selected the thermodynamically stable ones. The relatively low H content compared to H pressure hydrides like LaH₁₀ is not surprising, given the much lower pressures.

2. There have been many theoretical calculations supporting Lu-N-H system with the lower superconducting transition temperatures, but their research perspectives and methods are different. It is recommended that the authors summarize these studies appropriately.

A: We thank the referee for his/her comment. Indeed, in the revised version of the manuscript, we have discussed the results of the newest pre-prints/articles available in the literature, pointing out the main methodological differences and comparing the conclusions.

3. Could the authors discuss more the similarities between Lu_2NH_3 , TMD, and MgB_2 ?

A: As required by the Referee, we have now included a longer discussion of the main similarities between Lu_2NH_3 , TMD, and MgB_2 .

References

- [1] N. Dasenbrock-Gammon, E. Snider, R. McBride, H. Pasan, D. Durkee, N. Khalvashi-Sutter, S. Munasinghe, S. E. Dissanayake, K. V. Lawler, A. Salamat, *et al.*, "Evidence of near-ambient superconductivity in a n-doped lutetium hydride," *Nature*, vol. 615, no. 7951, pp. 244–250, 2023.
- [2] D. R. Hamann, "Optimized norm-conserving vanderbilt pseudopotentials," *Phys. Rev. B*, vol. 88, p. 085117, Aug 2013.
- [3] M. Schlipf and F. Gygi, "Optimization algorithm for the generation of oncv pseudopotentials," *Computer Physics Communications*, vol. 196, pp. 36–44, 2015.
- [4] N. S. Pavlov, I. R. Shein, K. S. Pervakov, V. M. Pudalov, and I. A. Nekrasov, "Anatomy of the band structure of the newest apparent near-ambient superconductor $\text{Lu}_{3-x}\text{N}_x$," *arXiv preprint arXiv:2306.09868*, 2023.
- [5] W. Wu, Z. Zeng, and X. Wang, "Investigations of pressurized lu-nh materials by using the hybrid functional," *arXiv preprint arXiv:2306.11511*, 2023.
- [6] F. Aryasetiawan, M. Imada, A. Georges, G. Kotliar, S. Biermann, and A. I. Lichtenstein, "Frequency-dependent local interactions and low-energy effective models from electronic structure calculations," *Phys. Rev. B*, vol. 70, p. 195104, Nov 2004.
- [7] W. Sano, T. Koretsune, T. Tadano, R. Akashi, and R. Arita, "Effect of van hove singularities on high- T_c superconductivity in H_3S ," *Phys. Rev. B*, vol. 93, p. 094525, Mar 2016.
- [8] H. Lee, S. Ponc e, K. Bushick, S. Hajinazar, J. Lafuente-Bartolome, J. Leveillee, C. Lian, F. Macheda, H. Paudyal, W. H. Sio, *et al.*, "Electron-phonon physics from first principles using the epw code," *arXiv preprint arXiv:2302.08085*, 2023.
- [9] K. P. Hilleke, X. Wang, D. Luo, N. Geng, B. Wang, and E. Zurek, "Structure, stability and superconductivity of n-doped lutetium hydrides at kbar pressures," *arXiv preprint arXiv:2303.15622*, 2023.
- [10] Z. Huo, D. Duan, T. Ma, Q. Jiang, Z. Zhang, D. An, F. Tian, and T. Cui, "First-principles study on the superconductivity of n-doped fcc- LuH_3 ," *arXiv preprint arXiv:2303.12575*, 2023.

REVIEWERS' COMMENTS

Reviewer #1 (Remarks to the Author):

The authors responded all the reviews raised by the referees and the revised manuscript seems suitable for publication in the journal of nature communications.

Reviewer #3 (Remarks to the Author):

I am satisfied with the efforts for revision by authors, thus recommending publication to the journal.

We want to thank the Referees for their efforts and time to review our work again, and are delighted to hear that all recommend publishing our manuscript in Nature Communications.